# Parathyroid Allotransplantation: A Systematic Review

**DOI:** 10.3390/medsci10010019

**Published:** 2022-03-15

**Authors:** Jaimie L. H. Zhang, Natasha M. Appelman-Dijkstra, Abbey Schepers

**Affiliations:** 1Department of Surgery, Leiden University Medical Center, 2333 Leiden, The Netherlands; a.schepers@lumc.nl; 2Department of Internal Medicine, Division Endocrinology, Leiden University Medical Center, 2333 Leiden, The Netherlands; n.m.appelman-dijkstra@lumc.nl

**Keywords:** hypoparathyroidism, parathyroid allotransplantation, immunosuppression, microencapsulation, macro-encapsulation, cell transplantation

## Abstract

**Background**: To date, there is no satisfactory treatment for patients with calcium and vitamin D supplementation refractive hypoparathyroidism. Parathyroid allotransplantation by design is a one-time cure through its restoration of the parathyroid function and, therefore, could be the solution. A systematic literature review is conducted in the present paper, with the aim of outlining the possibilities of parathyroid allotransplantation and to calculate its efficacy. Additionally, various transplantation characteristics are linked to success. **Methods:** This review is carried out according to the PRISMA statement and checklist. Relevant articles were searched for in medical databases with the most recent literature search performed on 9 December 2021. **Results:** In total, 24 articles involving 22 unique patient cohorts were identified with 203 transplantations performed on 148 patients. Numerous types of (exploratory) interventions were carried out with virtually no protocols that were alike: there was the use of (non-) cryopreserved parathyroid tissue combined with direct transplantation or pretreatment using in vitro techniques, such as culturing cells and macro-/microencapsulation. The variability increased further when considering immunosuppression, graft histology, and donor–recipient compatibility, but this was found to be reported in its entirety by exception. As a result of the large heterogeneity among studies, we constructed our own criterium for transplantation success. With only the studies eligible for our assessment, the pooled success rate for parathyroid allotransplantation emerged to be 46% (13/28 transplantations) with a median follow-up duration of 12 months (Q1–Q3: 8–24 months). **Conclusions:** Manifold possibilities have been explored around parathyroid allotransplantation but are presented as a double-edged sword due to high clinical diverseness, low expertise in carrying out the procedure, and unsatisfactory study quality. Transplantations carried out with permanent immunosuppression seem to be the most promising, but, in its current state, little could be said about the treatment efficacy with a high quality of evidence. Of foremost importance in pursuing the answer whether parathyroid allotransplantation is a suitable treatment for hypoparathyroidism, a standardized definition of transplantation success must be established with a high-quality trial.

## 1. Introduction

Hypoparathyroidism (hypoPT) is a frequently occurring complication of surgery in the central neck compartment due to damage or avascularization of the parathyroids (~8% in the U.S.) [1]. HypoPT or aparathyroidism causes hypocalcemia and, when proven permanent, it results in the need for lifelong calcium and vitamin D supplementation. However, the issue with supplements is the frequent lack of their sufficiency leading to symptomatic hypocalcemia, such as muscle cramps [2], which can escalate to life-threatening hospital admissions, but also the difficult balance between controlling the symptoms and supplement-induced side effects, including gastrointestinal complaints and renal calcifications.

More advanced manners of managing hypoPT are recombinant human PTH (rhPTH)replacement therapy and parathyroid allotransplantation (PTx) [3,4,5]. The use of rhPTH was recently approved as an alternative therapy in the event of conventional treatment failure, since multiple articles demonstrated its success evidenced by the normalization of biochemical parameters, as well as a decrease in symptoms and supplementation [6,7,8,9]. In one study, for example, 16 of 33 (48%) participants were able to eliminate active D supplementation over a period of 6 years. Despite the promising results, its effect on quality of life has been varying; long-term open-label studies suggested an improvement [6,9], while, on the contrary, a short-term placebo-controlled trial did not [10]. Additionally, rhPTH replacement therapy has shown inconsistent effects on hypercalciuria [11]; the relation to renal calcifications or insufficiency is unclear [7], and no data are available concerning its potential long-term benefits or detriments, such as the risk of developing osteosarcoma [12]. Furthermore, therapy with rhPTH is accompanied by substantial costs: the incremental cost-effectiveness ratio of rhPTH was reported to be USD 804,378/QALY, as opposed to the standard supplementation in hypoPT patients reasonably managed by the latter [13].

These shortcomings might argue for the alternative treatment option of PTx. It has several benefits over the use of rhPTH replacement therapy; in theory, it is a one-time curative procedure and the cost of preparing cells is substantially low [14]. However, the main obstacle to success is the development of host inflammatory and immune responses against the graft, leading to graft failure [15]. To solve this problem, the literature describes various transplantation techniques to improve survival [16,17,18,19]. Although, to date, PTx is not a prominent topic nor widely available, further research in this area may lead to major improvements in graft survival, and thus a long-awaited definite treatment for chronic hypoPT.

The aim of this systematic review is to present an overview of methods used for PTx, with an analysis of long-term functional data to determine treatment efficacy. We additionally intend to identify the characteristics that are correlated with success.

## 2. Materials and Methods

### 2.1. Search Strategy and Study Selection

This systematic review was conducted according to the PRISMA guidelines and checklist. A systematic literature search was conducted to identify the relevant articles using PubMed, MEDLINE, Embase, Web of Science, COCHRANE Library, Emcare, and Academic Search Premier containing keywords, including free text words. The search strategy is shown in the Appendix A. The most recent literature search was performed on 9 December 2021. We only included original research articles for this search. The additional inclusion criteria were that the articles had to be published from the year 1980 and onwards, written in English, and contain the description of a protocol for human PTx with hypoPT of any cause, without involving fetal body materials.

JLHZ and AS independently evaluated the titles and abstracts of the studies with disagreements being resolved by consensus. The information in subsequent publications involving the same patient cohort was combined. The references of the articles included were checked for additional relevant articles.

### 2.2. Data Collection

Using data extraction templates, the following information was obtained in the full text of studies fulfilling the inclusion criteria: study characteristics (study design, number of male and/or female recipients, number of PTx performed, donor tissue histology and vital status, donor–recipient relatedness, ABO and HLA compatibility, and follow-up period (FU: defined as the time from PTx to the last documented moment of collecting the recipient data. In case of a failed graft, this was the time from PTx to the report of graft failure)); biochemical parameters; presence of hypocalcemic symptoms; use of supplementation; use of immune suppressants; transplantation methodology; success/survival rate; quality of life; adverse events; and mortality. Data collection was performed by JLHZ with control by the remaining authors.

### 2.3. Quality Assessment

The points of appraisal for the quality assessment were based on a tool developed for the evaluation of case reports/series [20]. The articles were evaluated according to the presence of information concerning the following: the absence of selection bias; clear definition of study population (were donor and recipient characteristics complete?); adequate ascertainment of hypoPT and graft function (were objective data stated?); sufficient FU duration for outcomes to occur (minimum of 1 month unless graft failure occurred); and reproducibility of the method. A total of 10 points could be awarded over the 5 categories rated, with “Yes” being worth 2 points, “Unclear” 1 point, and “No” 0 points. A score of >7 points was considered as high, 6–7 fair, and <6 low quality. The quality assessment was performed by JLHZ with control by the remaining authors.

## 3. Results

### 3.1. Literature Search and Study Characteristics

The literature search identified 992 articles, of which 423 were original. After applying the inclusion criteria to the title and abstract, 33 articles remained, for which a detailed assessment of the full text was performed. In the final 24 studies included, data extraction was performed (Figure 1). This number contained 3 articles by Tollockzo et al. [17,21,22] involving the same patient cohort, and thus a total of 22 unique patient cohorts were involved. The cross-reference check performed on the full-text articles did not provide any additional relevant articles. The key characteristics of included studies are presented in Table 1. Each study by design was prospective, with the sample size ranging from 1 to 85 PTx recipients. A total number of 203 transplantations was performed on 148 recipients. All the recipients were reported to have received supplementation prior to transplantation and, if information about preoperative hypocalcemic symptoms was available, this was always found to be present. In the cases where authors did not provide this information, the recipients were found to have calcium levels that were below normal. Multiple parathyroid allotransplantations were performed on 45 recipients, 2 on 45 recipients and 3 on 10 recipients. Quality of life and mortality were not mentioned in any study as an outcome in neither donor nor recipient.

### 3.2. Donor Characteristics, Compatibility, and Transplantation Technique

Parathyroid grafts were usually obtained from living donors, but also a heart-beating cadaver, braindead and deceased donor were used. The histology of the grafts varied from a healthy to hyperplastic origin, with the hyperplastic branch being divided further into primary and renal hyperparathyroidism (Cabane et al. [26] did not mention the graft histology of one donor). Healthy tissue was grafted less often (6 PTx), but, in the case of a related donor and recipient, this was the tissue histology (2/2 PTx). The donor–recipient compatibility details were often neglected in studies: ABO and HLA compatibility were reported in, respectively, 14 and 9 of the 22 total cohorts. Studies mentioning only “compatibility was searched for”, “donor and recipient were evaluated” or “noncompatible”, without specifying whether this statement meant ABO and/or HLA, were regarded as insufficient [26,32,41].

Observed variability in the PTx technique was the usage of (non-)cryopreserved parathyroid tissue combined with direct transplantation or one of the following in vitro techniques: the culturing of parathyroid cells, micro-encapsulation [26,38], and macro-encapsulation [33,37]. The cryopreservation of grafts was frequently utilized until the identification of a suitable recipient. Eleven manuscripts used cryopreservation, which was combined with all the in vitro techniques mentioned above. Nawrot et al. [17] performed transplantations with the cultured grafts that were cryopreserved the longest: repeat transplantations were performed with grafts stored for a minimum of 6 months. Not utilizing any in vitro technique meant mincing the parathyroid tissue, followed by the direct transplantation in a muscle group of choice. Culturing parathyroid cells was an in vitro technique applied with the aim of decreasing the immunogenicity of the parathyroid tissue. Micro- and macro-encapsulation were the in vitro methods employed in a less frequent manner but have the imperative advantage of potentially making immunosuppression unnecessary in transplant-naïve recipients (i.e., those without other organ transplants). In macro-encapsulation, many cells are placed in an immunoprotective device, whereas in micro-encapsulation, small clusters of cells are surrounded by a gel capsule. The goal of both approaches was the same: to protect the inner cells from the host immune system, while at the same time allowing for the bidirectional diffusion of oxygen, nutrients, and waste to support cell survival [42]. All the transplantations were grafted on an ectopic site; most often the forearm was used (87%, 177/203 PTx), followed by the deltoid (9%, 19 PTx). The rare one-time sites were the rectus abdominis, femoral artery, and omental tissue. Basic information like the graft location was not provided in some articles [26,34].

### 3.3. Immunosuppression

Of the studies using immunosuppressives (n = 16), 9 featured transplant-naïve patients. The immunosuppressive regimen of the transplant-naïve cohort was simpler and transient as solely a PTx is not viewed as outweighing the risks of immunosuppressives. Between these studies, six used both induction and maintenance therapy and two had either induction or maintenance. The remaining study, by Puig-Domingo et al. [35], mentioned the immunosuppressives that were used but not the dose nor the administration time frame. Most commonly, the immunosuppressive regimen consisted of a bolus induction dose of 250–500 mg i.v. methylprednisolone, followed by the gradual tapering of this dose throughout the first postoperative week (maintenance). The regimen seemed somewhat similar, but more than half of the studies of this specific patient group were published by the same research group in Turkey. Other groups often did not have multiple publications and were seen to add basiliximab, tacrolimus, or cyclosporin A. There were also outliers in immunosuppression duration: Hermosillo-Sandoval et al. [32] continued 2.5 mg/week of prednisone for 6 months and Agha et al. [23] administered the same dose with prednisolone instead, which was continued permanently.

Continuing with the transplant-exposed cohort requiring permanent immunosuppressives, two of six articles decided not to perform induction therapy. Great variation was observed in the induction as well as the maintenance immunosuppressive therapy. Between the research groups, the only common feature seemed to be using (methyl)prednisolone as an induction immunosuppressive. In one study, it was left at this, while in the remaining studies, antithymocyte globulin, azathioprin, basiliximab, cyclosporin, mycophenolate, tacrolimus, or a combination of these was added. Rahusen et al. [36] was the only article reporting adverse effects because of immunosuppressive drugs: cyclosporin toxicity and the moderately acute cellular rejection of the kidney was diagnosed on day 26 post-PTx, while the remaining organs remained fully functional. Rejection was treated and cyclosporin was converted to tacrolimus with no diagnosis of a second rejection in the following year. Some studies in this subgroup did not specify the immunosuppressive doses [27,28,30].

### 3.4. Efficacy of the Treatment

Authors frequently did not mention their definition of “graft survival" or "success”. Therefore, instead of relying on the studies’ (often absent) success criteria, we decided to make up our own criterion, bringing forth uniformity and a filter for quality. We defined transplantation success as a considerable reduction or stop in the need for supplements or hypocalcemic symptoms, combined with the serum calcium remaining in the normal range (2.15–2.55 mmol/L or 8.62–10.22 mg/dL). The criterion was drawn up to require information that was as minimal as possible, while preserving the clinical relevance. Fourteen studies provided sufficient information for our assessment. The pooled success rate of these studies was 46% (13/28), with a median FU duration of 12 months (Q1–Q3: 8–24 months). The FU duration of these studies varied considerably as this period ranged from 1 to 48 months. In the subgroup analysis, the pooled PTx success of, respectively, no, transient, and permanent immunosuppression were as follows: 25% (1/4) with a median FU of 10.5 months (Q1–Q3: 6.5–11.5), 47% (8/17) with a median FU of 24 months (Q1–Q3: 3.5–41.5), and 67% (4/6) with a median FU of 9.5 months (Q1–Q3: 8–18). In the no immunosuppression group, all the studies applied micro- or macro-encapsulation. The following pooled success rates were obtained for, respectively, fresh and cryopreserved tissue: 59% (10/17) with a median FU of 12 months (Q1–Q3: 8.5–24) and 30% with a median FU of 12 months (3/11) (Q1–Q3: 3–48). Notably, five out of six PTxs with permanent immunosuppression were also carried out using fresh grafts. The effect of donor–recipient compatibility on the success was not assessed because of information being frequently unavailable or unclear.

Instead of transplantation success, four studies stated their own criteria for endocrine function or graft survival; their criteria were less strict than ours, but sometimes composed of more compulsory components, such as an increase in PTH compared to pre-PTx, the detection of the graft by sestamibi, the detection of graft vascularization by ultrasonography, and the presence of a PTH gradient between blood samples taken from grafted and non-grafted arms [17,21,26,32]. In the study of Nawrot et al. [17], 85 patients underwent PTx without immunosuppressives and their mean allograft survival amounted to 20.7% (24/116) at 6.4 months. In Tolloczko et al. [21], 15 of the 23 transplantations performed showed endocrine activity after month 1 post-PTx; this number dropped to 4 at month 4, and ultimately 1 was left at month 12. This last case maintained PTH levels within a normal range for 14 months after PTx. The allograft function criteria were not stated by Aysan et al. [25], but they, nevertheless, did report the allograft functions in 7 of 10 patients at a median FU of 12 months (range: 9–15 months).

Another outcome of interest was the recipients remaining asymptomatic post-PTx for at least one year without any form of supplementation. This was achieved in three studies: Agha et al. [23], Rahusen et al. [36], and Yucesan et al. [40]. This period lasted, respectively, for 35, 24, and 12 months. All three cases involved immunosuppression and use of fresh tissue and two appeared to be ABO-matched (ABO unknown in Rahusen et al. [36]).

### 3.5. Repeat Transplantations

Multiple transplantations were performed on 45 recipients: 2 in 45 recipients and 3 in 10 recipients. When evaluating the repeat transplantations in respect to prior attempts, usually no considerable changes in the transplantation technique were made. Namely, the graft location and amount of tissue transplanted were both observed to have changed only once in respect to prior attempts. Repeat PTx was carried out with either cryopreserved tissue from the previous donor or from a new donor, sometimes also signifying a different parathyroid histology. These changes were low in frequency and had varying results. Thus, no patterns were discovered. Nawrot et al. [17] had 25 patients undergo re-transplantation and investigated if there might be a difference between the parathyroid tissue originating from the same or a new donor, but no association was found. In total, four recipients were subjected to a third PTx with cryopreserved grafts from unfamiliar donors and without immunosuppressives, but FU duration was short in all cases due to rejection or unsatisfactory increase in PTH [21,34].

### 3.6. Quality Assessment

The quality of studies is summarized in Table 2. Case reports are inherently vulnerable to selection bias and were graded accordingly. None of the studies containing more recipients stated having systematically included patients. Frequent reasons for deducting points under the category of “clear definition of study population” were the absence of information about basic transplantation compatibility, such as relatedness and donor-recipient compatibility (ABO and HLA match). Other frequently recurring reasons for degradation were insufficient reproducibility due to the unavailability of information regarding a detailed immunosuppressive drug regimen or the quantity of parathyroid tissue transplanted. Only 6 of 22 studies were rated to have a high overall quality. Four of six studies rated with a high overall quality had sufficient clinical outcome data to be assessed by our success criterion mentioned previously.

## 4. Discussion

This systematic review was carried out with the aim of presenting an overview of the methods used for PTx, plus an analysis of the long-term functional data to determine treatment efficacy and favorable transplantation characteristics. From this, it emerged that manifold possibilities concerning the modality of PTx were explored: varying methods of tissue preparation, transplantation technique, use of (transient) immunosuppression, and micro- and macro-encapsulation. The dissimilarity between studies increased further when including variables such as ABO and HLA match, histologic origin of the graft, donor vital status, and re-transplantation with (un)familiar donor(s). Based on the literature reviewed, the PTx success rate according to our own definition of transplantation success was calculated to be 48% with a median FU of 12 months (Q1–Q3: 8–24 months). However, the level of evidence is scarce due to the unsatisfactory quality of articles.

Elaborating on the quality of studies, first and foremost, there is a high suspicion of the presence of publication and selection bias. Most articles reviewed were case reports and among case series, at best, a description of the selection criteria for recipients was provided. A glaring and frequently recurring obstacle was the absence of essential information when dealing with transplantations. Moreover, there was generally low expertise in carrying out the procedure based on number of publications, and there was no consensus in the definition of transplantation survival nor success. This, together with the diversity of studies, made it increasingly difficult to synthesize clinical recommendations. Because of this, we constructed our own success criterion aimed to filter for quality and to promote uniformity in the definition of transplantation success. We hoped this would present a more realistic representation of the treatment efficacy in comparison to other systematic reviews on the same topic that did not set a filter [43,44]. As expected, our calculated success was decreased in respect to that of Kim et al. [43]: we calculated a 47% success rate with a median FU of 24 months as supposed to their 73.3% with a mean FU duration of 18.6 ± 14.3 months for transient immunosuppression, and our 67% success rate with a median FU of 9.5 months as supposed to their 87.5% survival with a median FU of 11.5 months for permanent immunosuppression (Parameswaran et al. [44] had an incomparable outcome). In summary, the suspected biases combined with the neglect of providing essential information about transplantation characteristics lead to the suspicion of the calculated success rate, most likely still overestimating the true effectiveness of this treatment modality in its current state.

The effect of the histologic origin of parathyroid tissue on transplantation success remains inconclusive. As the adage goes: “you never develop chronic hypoparathyroidism with auto-implantation of parathyroid tissue” [45,46,47]. As time progressed, evidence showed this statement to be built on a weak foundation [48,49,50,51,52]. We, however, do speculate on the usage of tertiary hyperparathyroidism-derived grafts due to the fact that an autonomously secreting nature could provide greater success in PTx. Further research has yet to prove this concept. One concern to keep in mind is the possibility of post-PTx hyperparathyroidism, but among the articles reviewed, this outcome was not reported. Moreover, whether extra caution should be taken in case the donor has high-setpoint calcium remains unknown as none of the articles utilizing PHPT or THPT-derived grafts reported pretransplant donor calcium levels.

The major limitations of this systematic review are publication bias and the fact that low-quality reports cannot produce a high-quality review despite painstaking effort. Nonetheless, we believe this manuscript presents a thorough update of those that are available on this topic and underlines the areas that need more attention. Although there is still much to be gained, Stevenson et al. [2] found the perspective of hypocalcemic patients in carrying out a clinical trial exploring the effectiveness of PTx to be positive. Subsequently, the same research group proposed a randomized trial investigating the need and extent of immunosuppression in PTx with extensive data collection [53].

In theory, PTx could provide a permanent cure to hypoPT, but, in the present state, re-transplantation is indicated because of the generally short-lived results. With this statement it must be recognized that re-transplantation has not yet been proven to be worthwile. Even so, PTx positively does seem to be feasible and safe for repeated application. The current evidence from the literature regarding the optimal strategy for PTx and its long-term benefits and harms is scarce due to the diversity and unsatisfactory quality of studies, but it seems to be more valid when the recipient receives permanent immunosuppression and a fresh graft. Indispensable for future research is the establishment of a standardized definition of transplantation success containing clinically relevant endpoints, enabling for uniform interpretation and analysis between studies. Furthermore, they should also be more thorough in reporting detailed patient-donor information. As to achieve a functional transplantation program, a high-quality prospective trial should be initiated comparing PTx with standard care.

## Figures and Tables

**Figure 1 medsci-10-00019-f001:**
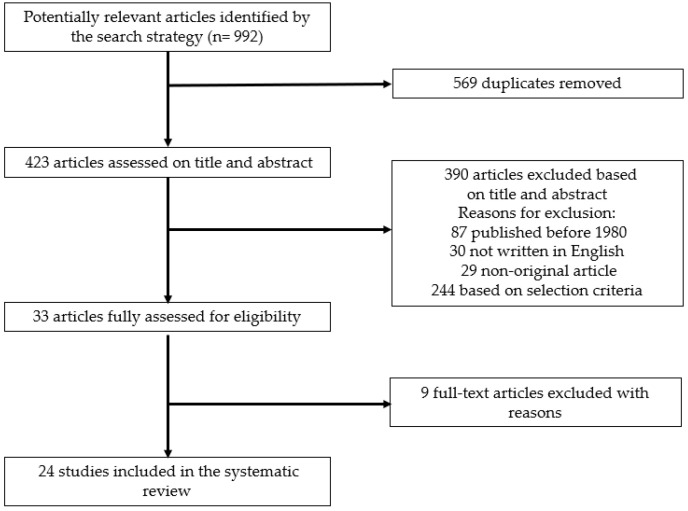
Study selection.

**Table 1 medsci-10-00019-t001:** Summary of study characteristics.

Study	# of Recipients (# of PTx)	Donor Information	Permanent/ Transient Immunosuppression	Transplantation Technique	Follow-Up Duration and Results	Success *
**Agha 2016** [23]	1 (1)	Living Related Normal histology ABO-compatible HLA N.A.	Transient: Basiliximab Prednisolone Tacrolimus	Minced parathyroid in the forearm	35 months Symptoms: absent Supplements: absent Calcium: 9.2 mg/dL PTH: 21 pg/mL	1/1 (100%)
**Alfrey 1992** [24]	1 (1)	Living Unrelated RHPT ABO N.A. HLA A2-matched	Permanent: Azathioprin Prednisone	Minced parathyroid in the forearm	13 years Symptoms: N.A. Supplements: N.A. Calcium: 8.6 mg/dL N-terminal PTH in grafted arm: 79 pmol/L	Assessment not possible
**Aysan 2016** [25]	10 (11)	Living Relatedness N.A. RHPT 4/10 ABO-compatible (retransplant N.A.) HLA N.A.	Transient: Methylprednisolone Prednisolone	Cultured parathyroid cells in the deltoid	Median FU: 12 months (range, 9–15 months) Symptoms: N.A. Supplements: absent in 7/10 Calcium:8.05 ± 0.61 mg/dL (mean) PTH: 8.3 ± 8.45 pg/mL (mean) Survival: 8/11 (72.3%)	Assessment not possible
**Aysan 2016** [14]	1 (1)	Living Relatedness N.A. RHPT ABO-incompatible HLA N.A.	Transient: Prednisolone	Cultured parathyroid cells in the deltoid	5 months Symptoms: absent Supplements: absent Calcium: 7.6 mg/dL PTH: 6.7 ng/L	0/1 (0%)
**Cabane 2009** [26]	1 (2)	PTx 1: Living Relatedness N.A. RHPT Noncompatible (not specified) PTx 2: N.A.	None	PTx 1: Microencapsulated minced parathyroid in the forearm PTx 2: Cryopreserved microencapsulated minced parathyroid in the leg	21 months (~11 months after second PTx) Symptoms: absent until month 3 Supplements: reduction Calcium: ~6.5 mg/dL PTH: ~1 pg/mL	0/2 (0%)
**Chapelle 2009** [27]	1 (1)	Braindead Relatedness N.A. Normal histology ABO N.A. HLA-B and HLA-DR match	Absent of detailed regimen Permanent: Cyclosporin Mycophenolate mofetil Prednisolone	Minced parathyroid in the forearm	13 months Symptoms: N.A. Supplements: absent Calcium: N.A. PTH: ~15 pg/mL	Assessment not possible
**Duarte 1985** [28]	1 (2)	PTx 1: Deceased Relatedness N.A. Normal histology ABO-compatible HLA-mismatched PTx 2: Living Relatedness N.A. PHPT ABO-compatible HLA-mismatched	Absent of detailed regimen Permanent: Azathioprin Prednisone	Cultured parathyroid cells in the forearm	28 months (10 months after second PTx) Symptoms: N.A. Supplements: reduction Calcium: 7.8–8.5 mg/dL PTH: 4.48 microliter Eq/mL (Biochemical values had an unclear time-point after second PTx)	1/2 (50%)
**Flechner 2010** [29]	1 (2)	PTx 1: Unrelated RHPT ABO-compatible HLA-A2, HLA-B62, HLA-Cw16, HLA-Dr11, and HLA-Dq match PTx 2: Same donor	Permanent: Mycophenolate mofetil Tacrolimus Prednisone Transient PTx 1: Mycophenolate mofetil Tacrolimus Prednisone Transient PTx 2: Methylprednisolone	PTx 1: Minced parathyroid in the forearm PTx 2: Cryopreserved minced parathyroid in the forearm	11 months (8 after second PTx) Symptoms: absent Supplements: reduction Calcium: 9.3 mg/dL PTH: 15 pg/mL	1/2 (50%)
**Garcia-Roca 2016** [30]	1 (1)	Living Related Healthy histology ABO-compatible HLA-haploidentical	Absent of detailed regimen Permanent: Mycophenolic acid Tacrolimus Transient: Basiliximab Methylprednisolone	Minced parathyroid in the rectus abdominis muscle	9 months Symptoms: absent Supplements: reduction Calcium: ~8 mmol/L PTH: 29 pg/mL	1/1 (100%)
**Goncu 2020** [31]	1 (1)	Living Unrelated RHPT ABO-compatible HLA-A match	Transient: Methylprednisolone	Cryopreserved cultured parathyroid cells in the omentum	1 year Symptoms: reduced Supplements: reduced Calcium: normal range (not specified) PTH: 1 pg/mL	1/1 (100%)
**Hermosillo-Sandoval** **2015** [32]	5 (5)	Living Unrelated PHPT HLA N.A. ABO N.A.	Transient: Cyclosporin Methylprednisolone Prednisone	Minced parathyroid in the forearm	2 years Symptoms: absent Supplements: reduction Calcium: 8.51 ± 0.1 mg/dl (mean) PTH: 19.07 ± 6.46 (mean) Survival according to author: 4/5 (80%)	4/5 (80%)
**Khryshchanovich 2016** [33]	1 (1)	Living Unrelated RHPT ABO-incompatible HLA mismatch	None	Cryopreserved macro-encapsulated parathyroid cells in the femoral artery	3 months Symptoms: absent, 1 event preceded by dyspepsia Supplements: reduction Calcium: ~1.7 mmol/L PTH: 21.15 pg/mL	0/1 (0%)
**Kunori 1991** [34]	1 (3)	PTx 1: Living Unrelated PHPT ABO N.A. HLA-DQw3 match PTx 2: Same donor PTx 3: Living Unrelated Healthy histology ABO N.A. HLA-A24 match	Transient PTx 1: Methylprednisolone PTx 2 and 3: None	PTx 1: Minced parathyroid in the deltoid PTx 2: Cryopreserved minced parathyroid with unknown transplantation location PTx 3: Minced parathyroid in the arm (not specified)	~8 months (~7 and 6 months from, respectively, the second and third PTx) Symptoms: infrequent Supplements: reduction Calcium: ~7 mg/dL PTH-C: ~0.1 ng/mL	0/3 (0%)
**Nawrot** **2007** [17]	85 (116)	Living Unrelated RHPT ABO-compatible HLA N.A.	None	Cryopreserved parathyroid cells in the forearm	6.35 ± 13.08 months Presence of symptoms and biochemical values were part of criteria for allograft survival. Graft failure required supplementation. Survival according to author: 20.7% (24/116)	Assessment not possible
**Puig-Domingo 2008** [35]	1 (2)	PTx 1: Living Relatedness N.A. RHPT ABO-compatible HLA-N.A. PTx 2: Same donor	Absent of detailed regimen Transient PTx 1: Everolimus Methylprednisolone PTx 2: Everolimus Methylprednisolone Mycophenolate	PTx 1: Cultured parathyroid cells in the forearm PTx2: Cryopreserved parathyroid cells in the forearm	~5 months (7 days after second PTx, graft failed) Symptoms: N.A. Supplements: N.A. Calcium: transiently normal after second PTx (graph) PTH: N.A.	Assessment not possible
**Rahusen 1997** [36]	1 (1)	Heart-beating cadaver Relatedness N.A. Healthy ABO N.A. HLA N.A.	Permanent: Antithymocyte globulin Azathioprin Cyclosporin Cyclosporin was converted to tacrolimus Transient: Methylprednisolone	Minced parathyroid in the forearm	2 years Symptoms: N.A. Supplements: absent Calcium: 8.6 mg/dL PTH: 19.1 pg/mL	1/1 (100%)
**Tibell 2001** [19]	4 (4)	Living Relatedness N.A. PHPT (donors 1 and 2) Hyperplastic parathyroid tissue (donors 3 and 4) ABO-compatible HLA mismatch	None	Macro-encapsulated minced parathyroid in the forearm	Small device: 4 weeks Large device: 8.5–14 months Symptoms: N.A. Supplements: reduction Calcium: N.A. PTH: N.A.	Assessment not possible
**Tolloczko 1996** [17,21,22]	23 (40)	Living Unrelated Secondary HPT (unclear if renal cause) ABO N.A. HLA N.A.	None	Cryopreserved cultured parathyroid cells in the forearm	9–34 months Biochemical values and supplementation were part of criteria for allograft survival. Survival according to author: 4.3% (1/23) at 14 months	Assessment not possible
**Torregrosa 2005** [37]	1 (1)	Living Relatedness N.A. RHPT ABO-compatible HLA-A3, HLA-B35, HLA-DR4 match	Permanent: Azathioprin Corticoid Cyclosporin Transient: Methylprednisolone	Minced parathyroid in the forearm	2 years Symptoms: N.A. Supplements: absent Calcium: N.A. PTH: 34 pg/mL	Assessment not possible
**Yucesan** **2019** [38]	1 (1)	Living Relatedness N.A. RHPT ABO N.A. HLA N.A.	None	Cryopreserved microencapsulated parathyroid cells in the omental tissue	1 year Symptoms: N.A. Supplements: absent Calcium: 7.2–9.1 mg/dL PTH: ~8 pg/mL	1/1 (100%)
**Yucesan** **2019** [39]	4 (4)	Living Relatedness N.A. RHPT ABO-compatible Recipient 1: HLA-DQB1 full match Recipient 2: HLA-A allele match Recipient 3: HLA-B allele match Recipient 4: HLA-DQB1 allele match	Transient: Methylprednisolone	Cryopreserved cultured parathyroid cells in the deltoid	4 years Symptoms: absent except for Recipient 3 Supplements: absent in Recipient 1 and 2, Recipient 3 reduction, and Recipient 4 returned to pre-transplantation levels after 3 years Calcium: Recipient 1: 9.2 mg/dL Recipient 2: 7.5 mg/dL Recipient 3: 8.9 mg/dL Recipient 4: N.A. (11.8 mg/dL at year 3) PTH: Recipient 1: 42 pg /mL Recipient 2: 14.4 pg/mL Recipient 3: 0.5 pg/mL Recipient 4: N.A. (2.5 pg/mL at year 3)	1/4 (25%)
**Yucesan** **2017** [40]	2 (2)	Living Relatedness N.A. HLA N.A. RHPT Donor 1: ABO-compatible Donor 2: ABO-incompatible	Transient: Methylprednisolone Prednisolone	Minced parathyroid tissue in the deltoid	1 year Symptoms: absent Supplements: absent Calcium: Recipient 1: 9.8 mg/dL Recipient 2: 7.4 mg/dL PTH: Recipient 1: 6.6 pg/mL Recipient 2:10.2 pg/mL	1/2 (50%)

* Success was assessed according to our criteria (considerable reduction or stop in the need for supplements or hypocalcemic symptoms, combined with serum calcium remaining in the normal range) at the end of FU. N.A.: not available, PTH: parathormone, and RHPT: renal hyperparathyroidism.

**Table 2 medsci-10-00019-t002:** Quality assessment of studies.

Reference	Absence of Selection Bias	Clear Definition of Study Population	Adequate Ascertainment	Sufficient Follow-Up	Reproducible	Overall Quality
**Agha 2016** [23]	No	Yes	Yes	Yes	Yes	High
**Alfrey 1992** [24]	No	Unclear	Yes	Yes	Yes	Fair
**Aysan 2016** [25]	No	Yes	Yes	Yes	Yes	High
**Aysan 2016** [14]	No	Yes	Yes	Yes	Yes	High
**Cabane 2009** [26]	No	No	Yes	Yes	Yes	Fair
**Chapelle 2009** [27]	No	Yes	Unclear	Yes	No	Low
**Duarte 1985** [28]	No	Yes	Yes	Yes	No	Fair
**Flechner 2010** [29]	No	Yes	Yes	Yes	No	Fair
**Garcia-Roca 2016** [30]	No	Yes	Yes	Yes	No	Fair
**Goncu 2020** [31]	No	Yes	Yes	Yes	Unclear	Fair
**Hermosillo-Sandoval 2015** [32]	No	Yes	Yes	Yes	Yes	High
**Khryshchanovich 2016** [33]	No	Yes	Yes	Yes	Unclear	Fair
**Kunori 1991** [34]	No	Yes	Yes	Yes	No	Fair
**Nawrot 2007** [17]	No	Yes	Yes	Yes	Yes	High
**Puig-Domingo 2008** [35]	No	Yes	Unclear	Yes	No	Low
**Rahusen 1997** [36]	No	Unclear	Yes	Yes	Yes	Fair
**Tibell 2001** [19]	No	Yes	No	Yes	No	Low
**Tolloczko 1996** [17,21,22]	No	Yes	Yes	Yes	Unclear	Fair
**Torregrosa 2005** [37]	No	Yes	Yes	Yes	No	Fair
**Yucesan 2019** [38]	No	Unclear	Yes	Yes	Yes	Fair
**Yucesan 2019** [39]	No	Yes	Yes	Yes	Unclear	Fair
**Yucesan 2017** [40]	No	Yes	Yes	Yes	Yes	High

## Data Availability

Available on reasonable request.

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
