# Peer review of "Parathyroid Allotransplantation: A Systematic Review"

_medsci, 2022, doi:10.3390/medsci10010019_

Round 1
Reviewer 1 Report
In the current systemic review, the authors highlight the efficacy regarding parathyroid allotransplantation. The review is well written and has an original and interesting central idea. However, some points deserve to be improved.
-
It would be wise to clearly explain in the introduction section the epidemiological side of the pathologies that require this type of transplantation, and also the explain the pathophysiology of parathyroid.
2. It is important to compare the method reported in this review with the results of other treatment methods.
3. My major remark is that the data collected presents a great heterogeity from a relatively reduced number of studies (n=24), this hinders drawing a solid conclusion and gives a rather descriptive aspect of the review.
Author Response
First of all, we the authors would like to sincerely thank you for reviewing our article. See below for our point-by-point response to your comments.
- We first would like to mention that we aimed to keep our systematic review concise. Despite this, the resultant review is not the shortest. As a response to your first point, we added a percentage to give the reader a clearer idea of the problem. Furthermore, we added a bit of pathophysiology in our introduction by stating that hypoparathyroidism occurs through devascularization of parathyroids. This hopefully helps readers who are not that familiar with this subject to understand it better
- In the discussion we mentioned the outcomes of a systematic review on the same topic (reference 47 and 48). Parameswaran had different outcomes and thus were not comparable.
- Indeed, the heterogeneity hindered drawing a solid conclusion. However, we still strived to make a conclusion with caution because we did not want to leave the reader with "more qualitative research was needed" as an answer. Our conclusion hopefully gives scientists a rough direction to which they should focus.
If you feel that we have not paid enough attention to your valuable feedback with the above, we would like to hear from you
Reviewer 2 Report
Zhang and colleagues performed a systemic review on case reports or small series of parathyroid allotransplantation. Although there are similar reviews published recently, the authors independently assessed the quality of source papers. The manuscript is well written. I would support the publication of this article.
(1) At least two very similar reviews have been published: reference #47 (Kim et al, Am J Surg) and DOI:10.1016/j.surge.2020.06.008 (Parameswaran et al, Surgeon 2021). The authors should compare their results with these two reviews and discuss the differences.
(2) The authors may cite a good updated review on this topic, DOI:10.1530/EJE-20-1367 (Mihai et al, Eur J Endocrinol 2021).
(3) Six of 22 studies were considered to have a high overall quality. Success was observed in 6/9 of patients among these high-quality studies. It may serve as a sensitivity analysis to corroborate the main finding.
(4) Spell out PRT (parathyroid?) in the main text.
Author Response
First of all, we the authors would like to sincerely thank you for reviewing our article. See below for our point-by-point- response to your comments.
- We first would like to mention that we aimed to keep our systematic review concise. Despite this, the resultant review is not the shortest. In the discussion we mentioned the outcomes of a systematic review on the same topic (reference 47 and 48). Parameswaran had different outcomes and thus were not comparable.
- As the article mentioned by you had a good proposal of a trial, we decided to cite this article.
- Because of the already lengthy review, we decided to not include a sensitivity analysis
- Thank you for bringing this up, this indeed increases readability
If you feel that we have not paid enough attention to your valuable feedback with the above, we would like to hear from you.